# Can machine learning methods be used for identification of at-risk neonates in low-resource settings? A prospective cohort study

Babar S Hasan,[1] Zahra Hoodbhoy ,[2] Amna Khan,[2] Mariana Nogueira,[3] Bart Bijnens,[3,4] Devyani Chowdhury[5]

BSH and ZH are joint first authors.

[1]Department of Pediatrics and Child Health, Sindh Institute of Urology and Transplantation, Karachi, Pakistan
[2]Department of Pediatrics and Child Health, Aga Khan University, Karachi, Pakistan
[3]IDIBAPS, Barcelona, Spain
[4]ICREA, Barcelona, Spain
[5]The Children's Hospital of Philadelphia, Philadelphia, Pennsylvania, USA

**Correspondence to**
Dr Babar S Hasan;
drbabarhasan@yahoo.com

## ABSTRACT

**Introduction** Timely identification of at-risk neonates (ARNs) in the community is essential to reduce mortality in low-resource settings. Tools such as American Academy of Pediatrics pulse oximetry (POx) and WHO Young Infants Clinical Signs (WHOS) have high specificity but low sensitivity to identify ARNs. Our aim was assessing the value of POx and WHOS independently, in combination and with machine learning (ML) from clinical features, to detect ARNs in a low/middle-income country.

**Methods** This prospective cohort study was conducted in a periurban community in Pakistan. Eligible live births were screened using WHOS and POx along with clinical information regarding pregnancy and delivery. The enrolled neonates were followed for 4 weeks of life to assess the vital status. The predictive value to identify ARNs, of POx, WHOS and an ML model using maternal and neonatal clinical features, was assessed.

**Results** Of 1336 neonates, 68 (5%) had adverse outcomes, that is, sepsis (n=40, 59%), critical congenital heart disease (n=2, 3%), severe persistent pulmonary hypertension (n=1), hospitalisation (n=8, 12%) and death (n=17, 25%) assessed at 4 weeks of life. Specificity of POx and WHOS to independently identify ARNs was 99%, with sensitivity of 19% and 63%, respectively. Combining both improved sensitivity to 70%, keeping specificity at 98%. An ML model using clinical variables had 44% specificity and 76% sensitivity. A staged assessment, where WHOS, POx and ML are sequentially used for triage, increased sensitivity to 85%, keeping specificity 75%. Using ML (when WHOS and POx negative) for community follow-up detected the majority of ARNs.

**Conclusion** Classic screening, combined with ML, can help maximise identifying ARNs and could be embedded in low-resource clinical settings, thereby improving outcome. Sequential use of classic assessment and clinical ML identifies the most ARNs in the community, still optimising follow-up clinical care.

## WHAT IS ALREADY KNOWN ON THIS TOPIC

⇒ Two-thirds of the global neonatal deaths occur in low/middle-income countries, but tools such as WHO Young Infants Clinical Signs (WHOS) or pulse oximetry have poor sensitivity to identify sick newborns.

## WHAT THIS STUDY ADDS

⇒ This study proposes a novel approach that combines highly specific tests like pulse oximetry and WHOS with a sensitive machine learning model which uses clinical features in a sequential manner to identify at-risk neonates in the community.

## HOW THIS STUDY MIGHT AFFECT RESEARCH, PRACTICE OR POLICY

⇒ This approach may help front-line health workers in identifying at-risk neonates and by using the strengths of each test, being able to triage and manage these neonates more effectively in low-resource settings.

deaths are prematurity, birth asphyxia, sepsis and congenital anomalies, most of which are congenital heart diseases (CHDs).[2]

Newborn screening at few hours of life, for identification of at-risk newborns (ARNs), has been proposed as an effective tool to reduce NMR.[3] In high-income countries (HICs), a combination of antenatal ultrasound and newborn screening enables early detection, diagnosis and intervention to prevent death or disability. Newborn screening, using a few drops of blood for genetic, metabolic and endocrine disorders, and American Academy of Pediatrics (AAP) pulse oximetry (POx) for critical congenital heart defects (CCHDs) are widely implemented in HICs.[4]

A recent systematic review by Plana *et al* reported that the overall sensitivity of POx for the detection of CCHD was 76.3% (95% CI 69.5% to 82.0%) (low certainty of the evidence), while the specificity was 99.9% (95% CI 99.7% to 99.9%) (high certainty

## INTRODUCTION

With a neonatal mortality rate (NMR) of 42/1000 live births, Pakistan accounts for 7% of all global neonatal deaths and ranks third on the list among countries with highest NMR.[1] The top four causes of neonatal

of the evidence).[5] While POx is a highly specific tool to detect CCHD, it will detect other non-cardiac conditions associated with hypoxia, such as neonatal sepsis, hypothermia and persistent pulmonary hypertension (PPHN),[6 7] where it has demonstrated a high specificity (>99.9%) but low sensitivity (42%).[8] POx thus has shown a broader utility as a newborn wellness screen, with a good feasibility and acceptability for screening newborns in primary care centres in low/middle-income countries (LMICs).[9]

A complementary newborn wellness screening tool widely used in LMICs is the WHO Young Infants Clinical Signs Study Group (WHOS), which identifies seven signs including fever ≥38°C, hypothermia (temperature <35°C), convulsions, lethargy, poor feeding, chest indrawing and tachypnoea (respiratory rate >60 breaths/min), and has a sensitivity of 74% and specificity of 79% to predict severe illness.[10] Though WHOS and POx have shown only modest sensitivity, these are simple and highly specific screening tools which may be used to identify newborn conditions contributing to NMR in low-resource settings.

Given their moderate sensitivity, there remains a need to optimise the performance of these tools and to develop improved strategies by including the use of a wider set of clinical parameters for community-based triage of ARNs.[10] The aim of this study was to assess the value of POx and WHOS, used on their own, in combination and enriched with machine learning (ML) from clinical features, to develop a screening tool for community health workers (CHWs) to identify ARNs.

## METHODS
### Study design and participants
This was an ethics committee-approved single-group prospective cohort study conducted at Ibrahim Hyderi, a periurban settlement of approximately 100000 individuals on the southeast of Karachi, Pakistan, from 1 October 2020 to 31 March 2022. The main occupation of individuals in this area is fishing with 40% below the poverty line.[11] The NMR at this site ranges between 40 and 45/1000 live births.[11] The eligibility criterion was a single live birth where the mother planned to stay in the study catchment area for at least 4weeks after delivery.

### Sample size estimation
The sample size was estimated based on the primary outcomes of this study. Neonatal sepsis has been reported to be as high as 14%,[2] while the prevalence of CHD in Pakistan is stated to be around 0.9%.[12] With a 95% CI, power of 80% and absolute precision of 0.6%, a frequency of 0.9% generated the maximum sample size of 951 neonates. To account for 5% loss to follow-up, the sample was inflated to 1000 neonates.

### Procedures
The Department of Pediatrics and Child Health at the Aga Khan University maintains a demographic and health surveillance system at this site whereby all households are enumerated. Leveraging the surveillance data for identification of pregnant women, we used a non-probability convenience sampling approach where trained CHWs visited these households to screen eligible women and obtained written informed consent for the participation of their newborn in the research study. All live births were registered within 72 hours, where clinical information about the pregnancy and delivery was collected (refer to online supplemental table 1), while danger signs were assessed using WHOS[10] along with POx (using the Masimo Rad-8 pulse oximeter).

The oxygen saturation (SpO$_2$) was measured on the right hand and foot (30s each) once the device's signal quality index showed stable green bars for >10s. The SpO$_2$ readings were recorded and categorised into green, yellow and red zone as per AAP recommendations (refer to figure 1 and online supplemental figure 1).[13] Neonates who had a positive WHOS or fell in the yellow or red zone for POx were referred for further evaluation and echocardiography with follow-up by a physician at the primary healthcare centre (PHC). As per PHC protocol, all clinically sick neonates who were prescribed antibiotics were followed daily for the next 7days. All neonates were also followed up at 4weeks of life to assess the vital status of the child, and hospitalisation, if any, during this period. For all neonatal deaths, a verbal autopsy was performed using the WHO 2016 verbal autopsy instrument by trained research staff[14] and coded by a physician to ascertain cause of death.

### Outcomes
The primary outcome of this study was the identification of ARN, defined as:
▶ Early neonatal sepsis: confirmed by a physician after referral based on WHOS or other clinical indications.[10]
▶ CCHDs: confirmed by echocardiography (transposition of the great arteries, double outlet right ventricle, complex single ventricle physiology, tetralogy of Fallot, truncus arteriosus).[15]
▶ Other cardiac conditions, such as severe PPHN (bidirectional patent ductus arteriosus (PDA) and right ventricular dysfunction).
▶ Hospitalisation/death within the first 4weeks of life.

Neonates with isolated non-haemodynamically significant CHDs, like small ventricular septal defects and atrial septal defects,[16] were not considered as ARNs.

### Statistical analysis and ML
Conventional data analysis was performed using STATA V.16.4 bit. Data were reported as percentages and means±SD where appropriate.

For the ML, we used the XGBoost implementation of gradient boosted decision trees.[17] For the first model

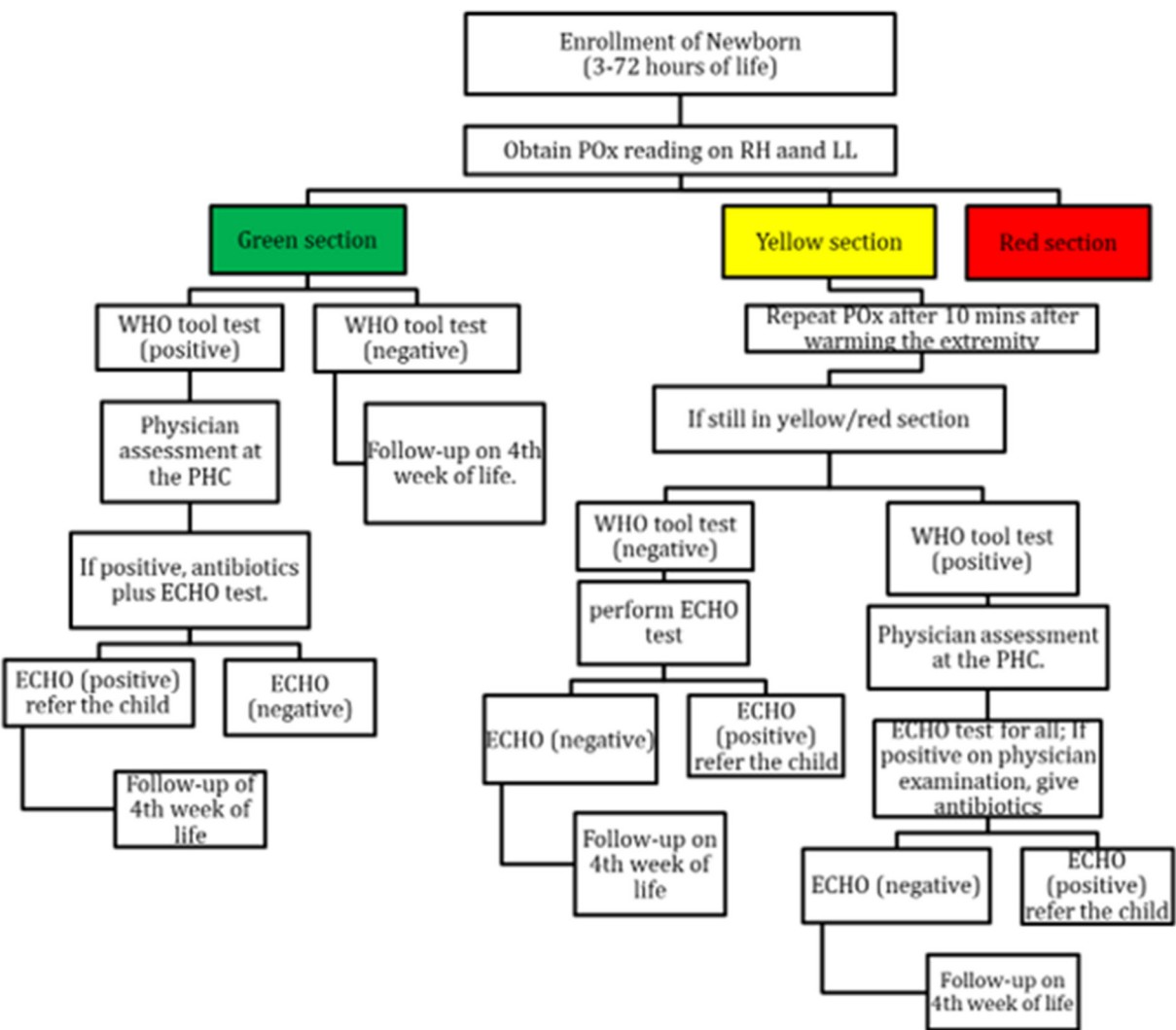

**Figure 1** Screening algorithm used by community health workers. Any child who failed the POx screening and/or WHO Signs by the community health worker at the home assessment was recommended an ECHO. ECHO, echocardiography; LL, lower limb; PHC, primary health centre; POx, pulse oximetry; RH, right hand.

(AllML), an extensive set of clinical variables (online supplemental table 1), the measurements obtained from POx and the outcome of WHOS (pass/fail) were used; for the second model (CliML), a smaller set of clinical variables selected based on a combination of prior expert knowledge on relevant features and the results of a model explainer, ranked variables according to importance for outcome prediction in AllML. We also used the SHapley Additive exPlanations (SHAP)[18] approach to understand the outputs of the ML models. The models were trained on 60% of the data and used threefold cross-validation while 40% were kept as test set. During the training stage, a grid search was performed to tune model hyperparameters, and those yielding the best cross-validation F1-score were selected. The F1-score and other performance metrics (sensitivity, specificity, precision, the receiver operating characteristic (ROC) and respective area under the curve) were then computed for the full training and the test sets.

Finally, a sequential use (SeqML) of WHOS and if negative followed by POx and if that too is negative, then CliML, was assessed. A ROC curve was constructed using a moving threshold on the probability provided by CliML and adding this to the (fixed) sensitivities and specificities of POx after WHOS.

## RESULTS

During the study period, 1336 neonates were enrolled from the community (figure 2). Maternal and neonatal characteristics of the study population are described in table 1. The mean maternal age was 27±5.1 years, 46% (n=618) did not have any formal education, 92% (n=933) had received at least one antenatal visit during the current pregnancy and one-third (n=397) delivered at home.

For the final outcome at 4-week follow-up, neonates were labelled as *well* or *ARN*. There were 19 neonates who were excluded from the analysis due to refusal for

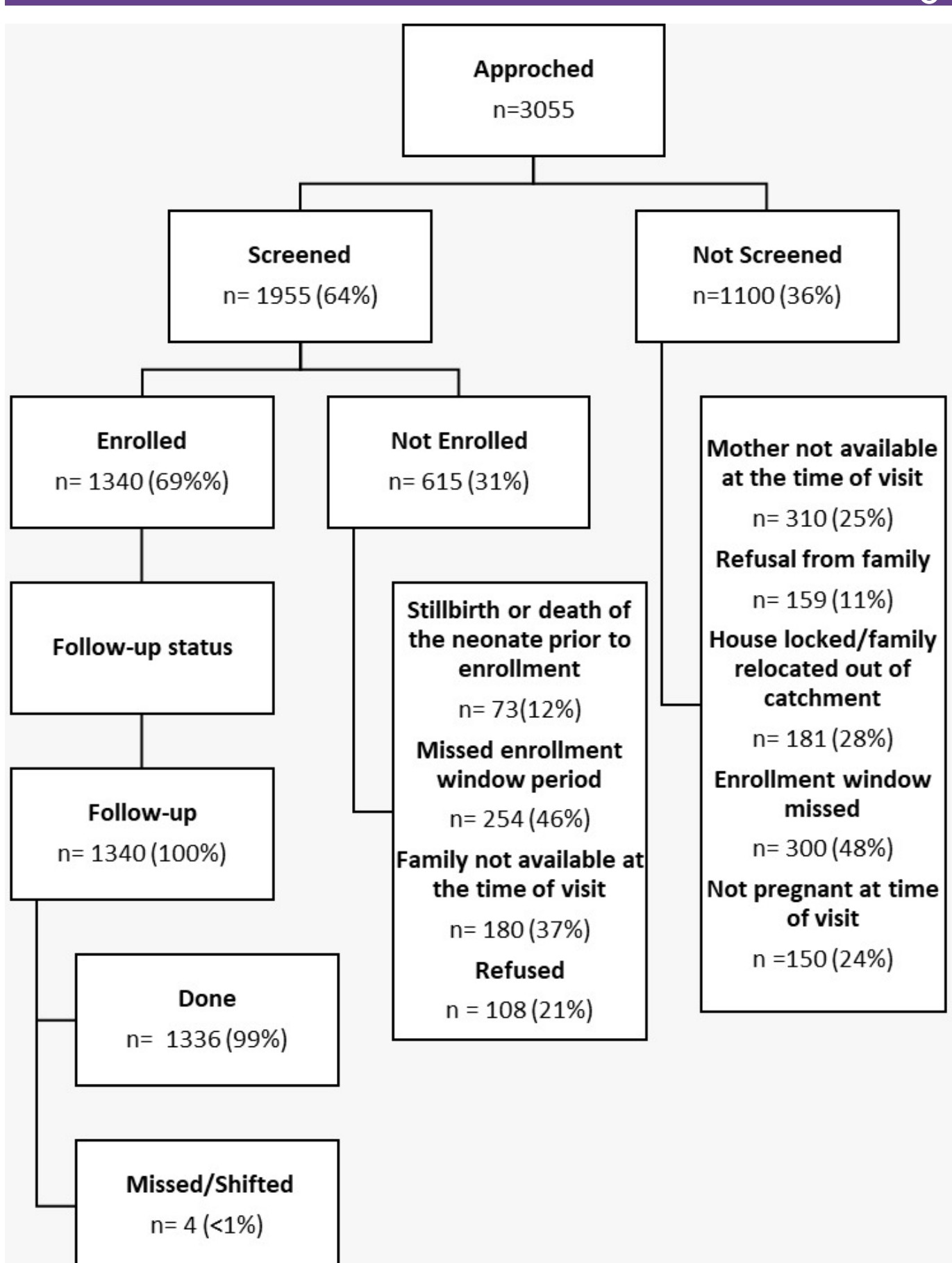

**Figure 2** Enrolment of study participants.

## Table 1  Baseline characteristics of mothers and neonates enrolled in the study

| Maternal characteristics (N=1336) | |
| --- | --- |
| Maternal age (in years), mean±SD | 27±5.1 |
| Literacy rate, n (%) | 618 (46) |
| Unemployed, n (%) | 1275 (95) |
| Number of pregnancies, median (IQR) | 3 (1–14) |
| Number of children alive, median (IQR) | 2 (0–10) |
| Received antenatal care in the current pregnancy, n (%) | 1248 (93) |
| Neonatal characteristics (N=1336) | |
| GA at delivery (in weeks), mean±SD | 39±4.5 |
| Preterm | 313 (23) |
| Term | 1023 (77) |
| Place of birth, n (%) | |
| Facility | 939 (70) |
| Home | 397 (30) |
| Mode of delivery, n (%) | |
| Spontaneous vaginal delivery | 1015 (76) |
| Assisted vaginal delivery | 3 (<1) |
| Caesarean section | 318 (24) |
| Gender, n (%) | |
| Female | 646 (48) |
| Male | 690 (52) |
| Hours at time of enrolment, median (IQR) | 48.2 (0–141) |
| <24 | 256 (19) |
| 24–48 | 462 (34.5) |
| 48–72 | 389 (29.1) |
| >72 | 229 (17.1) |
| Weight at enrolment in kg, median (IQR) | 2.84 (1.19–5.72) |
| Difference in hours between pulse oximetry screening and echocardiography | |
| <24 | 42 (60) |
| 24–72 | 14 (20) |
| >72 | 14 (20) |

GA, gestational age.

follow-up. We identified 68 of 1317 (5.1%) ARNs. These included 40 cases (59%) of neonatal sepsis, 2 cases (3%) of CCHD (both tetralogies of Fallot, one also with sepsis), 1 case (1.5%) of severe PPHN (bidirectional PDA with right ventricular dysfunction), 8 (11.5%) hospital admissions and 17 deaths (25%). The detailed characteristics of the study participants who died are described in online supplemental table 2. Of the neonatal deaths (n=17), 76% (n=13) passed both WHOS and POx screening. Five of these (30%) had the screening performed at less than 24 hours of life. Five were early neonatal deaths with approximately 2.5 days between enrolment and death, while for the 12 late neonatal deaths, the median

difference between enrolment and death was 16.5 days (IQR 6–26.5).

The distribution of the neonates according to WHOS and POx screening is shown in figure 3, with more details in online supplemental table 3.

### ARN screening performance

A total of 1317 neonates were included in the predictive modelling to assess ARN. The training set (60%) consisted of 790 neonates, while the test set included 527. Online supplemental table 4 summarises all results. As expected, WHOS (absent/present) and POx (pass/fail) showed a high specificity, but only moderate sensitivity to detect ARN. Combining both improved overall sensitivity as compared with the singular tests while preserving specificity. When comparing the results in the training and test set, they showed similar performance, confirming their stable performance in this population.

The ML model based on the clinical variables (CliML) showed good sensitivity/specificity on training, but much worse performance during testing, indicating overfitting.

AllML, trained on all assessed information at once, showed excellent performance at training, but reduced sensitivity in the test set indicating overfitting. The variables that contributed most to the decision included maternal features such as tobacco use, previous pregnancies, child deaths and pregnancy complication, as well as neonatal features, such as gender, gestational age at delivery and birth weight as indicated by the SHAP analysis (online supplemental figure 2).

To find a balance between the high specificity of WHOS and POx and the potentially higher sensitivity of CliML, a sequential testing approach, composed of first assessing WHOS, followed by POx and CliML to further identify ARN, was investigated (figure 4). Despite some overfitting during training, SeqML clearly showed increased sensitivity in the test set, as compared with all other approaches. Online supplemental figure 3 shows the ROC curve for both CliML as well as the sequential approach of SeqML.

### DISCUSSION

High NMR in LMICs is an important public health problem that can be addressed by timely detection of ARN. It is important to identify a community-based implementation framework where neonates can be appropriately followed by confirmatory testing and affordable management pathways.[19]

This study showed that a sequentially deployed strategy for identification of ARN can outperform the traditional approach of using only POx or WHOS. The model identifies a maximal number of ARNs and can be implemented in a staged manner where those failing WHOS and POx can be referred sooner to a higher-level care facility, while intermediate-risk neonates detected by the SeqML are monitored at home by a CHW.

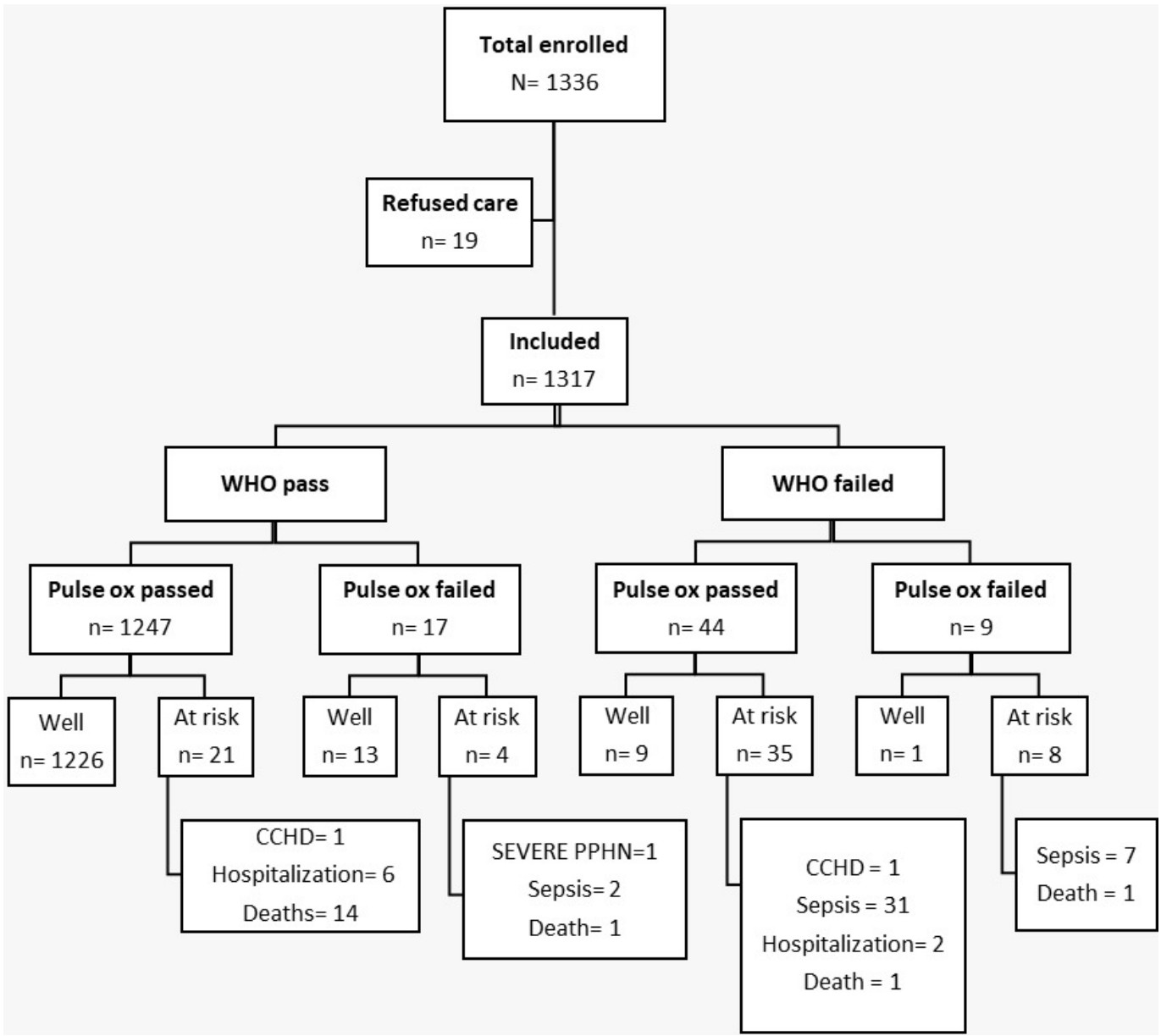

**Figure 3** Distribution of neonates who failed the WHO Signs and pulse oximetry (POx) screening. CCHD, critical congenital heart defect; PPHN, persistent pulmonary hypertension.

WHOS has modest sensitivity to detect non-infectious aetiologies. In the framework presented by the Global Initiative for Children's cardiac Surgery, POx and WHOS are a part of basic infrastructure for timely outpatient screening for cardiac disease.[20] POx is recommended for screening for CCHD and is also able to detect other hypoxia-causing non-cardiac conditions.[13 21] This has led to proposed recommendations of the WHO Integrated Management of Childhood Illnesses guidelines for detecting severe pneumonia to include $SpO_2$ of <93% as a point for referral.[22] In this study, POx is highly specific for identifying ARN, but has a poor sensitivity when used standalone. Among factors that might contribute to this is the timing of the screening as while the ductus arteriosus is still open, there is a higher sensitivity to detect life-threatening CCHD.[23 24] In the current study, there were four cases who were screened >24 hours after birth, did not have WHOS and passed the POx screening, but died within 7–14 days of birth at home where cause of death could not be ascertained. It may be possible that an underlying cardiac condition, especially left-sided lesions, may have been missed due to late neonatal screening. Additionally, the POx screening devices used only provided readings of the saturation values and heart rate and did not have visual feedback on acquired waveforms. While out of the scope of this manuscript, when examining the stored waveforms, it was observed that the quality of tracing might have also contributed to the lower sensitivity.

Besides the outcome of WHOS and POx for screening, the study showed that their combination allows to increase sensitivity while maintaining high specificity,

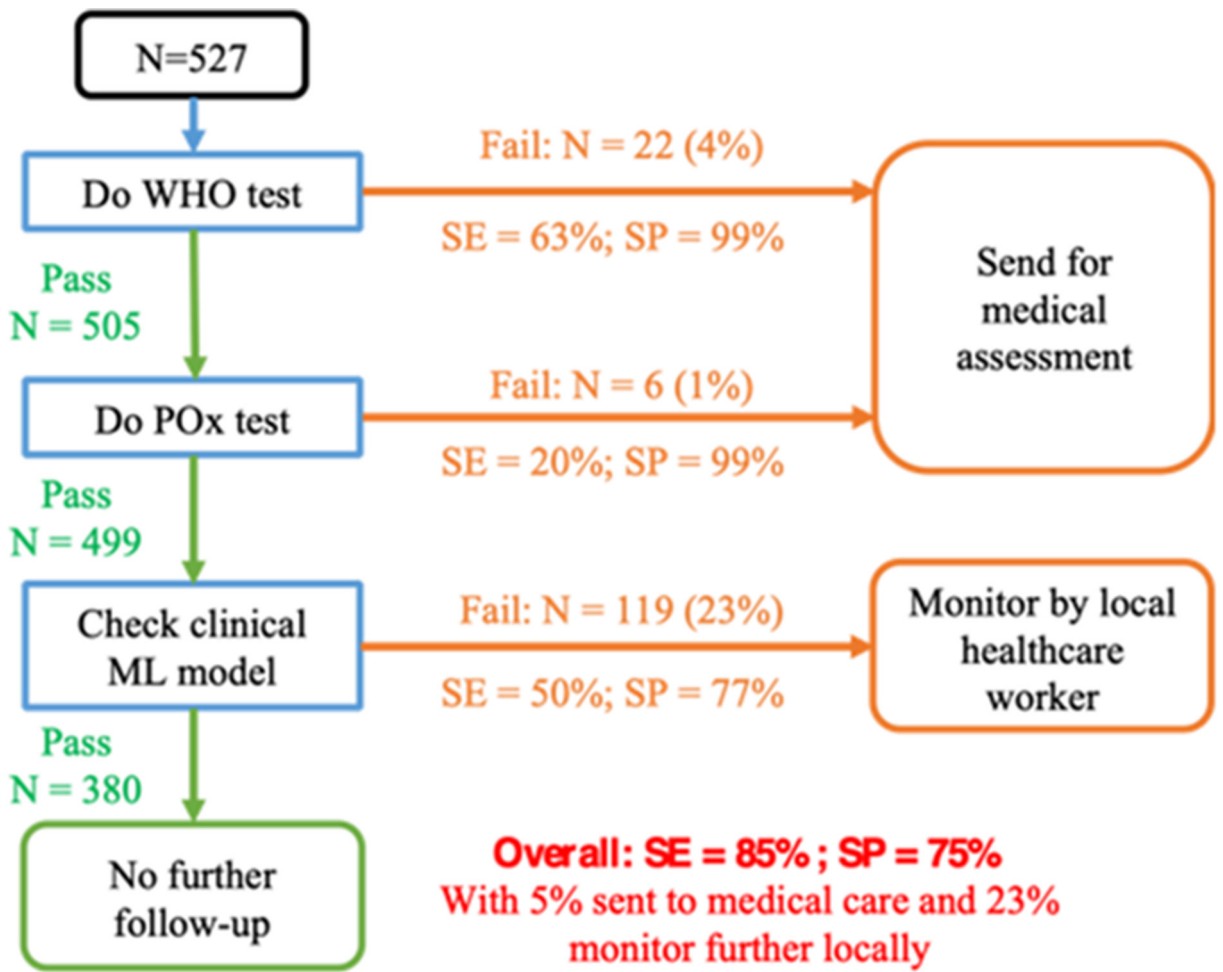

**Figure 4** Potential integration of the machine learning (ML) model in community-based workflow. POx, pulse oximetry; SE, sensitivity; SP, specificity.

indicating detecting other aetiologies associated with ARN. However, the modest sensitivity of the combination of both indicates that not all ARNs are identified. CliML provided predictive information, but only at modest sensitivity and specificity. Training a singular ML model (AllML) using all available information at once showed potential but had problems with generalisability given that the performance in the test data was inferior to the training. The sequential screening model (SeqML) where WHOS, a specific test that first identifies neonates at risk of sepsis and then identifies aetiologies associated with decreased saturation, through POx, and finally identifies those at higher risks using an ML algorithm based on clinical information, was most successful in identifying ARN. This approach optimises a staged follow-up given that WHOS and POx identify those that clearly need urgent medical attention, while the clinical data can be embedded and entered in an application on the CHW's mobile phone that identifies an increased risk without unique aetiology, but that could benefit from closer follow-up. The drawback of this sequential approach is the high number of false positives (23.6%, n=312). Rather than immediate referral of these cases, we propose that children who pass WHOS and POx, but fail CliML, should have regular follow-ups by the CHW as per the WHO guidelines for postnatal care and refer if needed (3, 7, 14 and 42 days of life).[25] During these follow-ups, the CHW can use WHOS and POx as tools to reassess the neonate and refer to a care facility if deterioration occurs. As seen in the simplified antibiotic therapy trial, home-based follow-up by CHWs at assigned days during the first 2 weeks of life helped with timely identification of sick neonates.[26] Further, the mother can also be counselled regarding danger signs in the neonate and the need to seek help immediately.[27] These approaches may be an effective strategy for task sharing and optimising resource utilisation without overburdening the health system with unnecessary referrals. This proposed data-driven and staged approach increases the sensitivity, while keeping the false positives manageable in the healthcare setting.

There are some limitations of the current study. The proposed clinical ML model needs to be externally validated in a larger population to assess its performance, given the difference between train and test performance. Additionally, the AllML model should be further explored and trained with larger datasets to assess its value as compared with the proposed sequential

approach. The complementary value of using the actual values of hand and foot $SpO_2$ together with heart rate, instead of the pass/fail information from the POx, may provide important information. After enrolment in the study, 73 of 1955 children died before any screening was performed, which may indicate a selection bias and may contribute towards the relatively low NMR (12.7/1000 live births) seen in this study. Also, the current model includes clinical parameters such as gestational age, which was calculated based on the self-reported date of last menstrual period which is known to have low reliability. Furthermore, POx was performed using one particular device only. Future research would require external validation of the sequential decision-making approach using the ML model in a community-based setting in improving detection of ARN.

## CONCLUSION

In this study, the value of different screening strategies to detect ARN was assessed. While assessing WHOS and performing POx are highly specific, adding an ML model based on clinical (maternal, neonatal and pregnancy-related) parameters could increase sensitivity. The increase in false positives would be easily managed if a staged, task-sharing approach involving community-based triage, close follow-up and early referral if needed by CHWs, is adopted. This model is also in alignment with the framework presented to decrease the global burden of childhood mortality due to non-communicable diseases.

**Acknowledgements** We would like to thank the study staff and the study participants for their contribution in this work. We would also like to acknowledge our partners including Masimo Corporation and our funders including Islamic Development Bank, Bill and Melinda Gates Foundation and the Fundació La Marató de TV.

**Contributors** BSH, ZH and DC designed and planned the study. BSH, ZH, AK and DC implemented the study and supervised data collection. BB, MN, AK and ZH conducted the analyses. BSH, ZH, AK, MN, BB and DC wrote and edited the manuscript. BSH is the guarantor of the overall content.

**Funding** This work has been funded by the Islamic Development Bank (10/HS/650 /441), the Bill and Melinda Gates Foundation (INV-021528), and the Fundació La Marató de TV3 (ref 202016-30-31).

**Competing interests** None declared.

**Patient and public involvement** Patients and/or the public were not involved in the design, or conduct, or reporting, or dissemination plans of this research.

**Patient consent for publication** Not required.

**Ethics approval** This study involves human participants and was approved by the Aga Khan University Ethics Review Committee (2019-0551-4976). Participants gave informed consent to participate in the study before taking part.

**Provenance and peer review** Not commissioned; externally peer reviewed.

**Data availability statement** Data are available upon reasonable request. Individual participant de-identified data will be made available upon request to the study authors 12 months after publication of this manuscript.

**ORCID iD**
Zahra Hoodbhoy http://orcid.org/0000-0002-0439-8293

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
