## [Reviewer comments · BMJ Paediatrics Open]

This paper was submitted to a another journal from BMJ but declined for publication following peer review. The authors addressed the reviewers' comments and submitted the revised paper to BMJ Paediatrics Open. The paper was subsequently accepted for publication at BMJ Paediatrics Open.

ARTICLE DETAILS

TITLE (PROVISIONAL)	Can machine learning methods be used for identification of at risk neonates in low resource settings? a prospective cohort study
AUTHORS	Hasan, Babar S. Hoodbhoy, Zahra Khan, Amna Nogueira, Mariana Bijnens, Bart Chowdhury, Devyani

VERSION 1 - REVIEW

REVIEWER	Dr. Peter Flom Peter Flom Consulting
REVIEW RETURNED	02-Jul-2023

GENERAL COMMENTS	I confine my remarks to statistical and methodological aspects of this paper. These were very well done. My only issue is that there needs to be more explanation of the figures regarding SHAP analysis. Peter Flom
---

REVIEWER	Dr. Imteyaz A. Khan Saint Peter's University Hospital, Pediatrics- NICU
REVIEW RETURNED	13-Jul-2023

GENERAL COMMENTS	There has been a great deal of planning and execution put into this prospective cohort study. It addresses a crucial question about how to screen at-risk newborns cost-effectively to prevent bad outcomes and decrease neonatal mortality. In Table S3, I would like to know what the echocardiography findings were for the two patients who had false negatives test. The sensitivity and specificity of pulse oximetry for detecting neonatal sepsis and congenital heart disease are shown in Table S3. The oxygen saturation of neonates is affected by neonatal sepsis in the late stages. Pulse oximetry is not an effective tool for screening and early detection of neonatal sepsis. Consequently, it makes the impression that an excellent tool is being used to detect something that isn't intended, and its usefulness is being questioned. Abbreviating single words such as sensitivity (SE) and specificity (SP) does little to condense the text, but rather makes quick
--

	understanding difficult and train of thought bumpy. I would like to know the method by which the study participants were selected. Was there any randomization done in order to prevent selection bias? My statistical colleague will handle the statistical portion of the manuscript, but I have serious doubts about pulse oximetry's 42% sensitivity. Pulse oximetry sensitivity for diagnosing non-cardiac pathology should be clearly stated as diagnosing of non-cardiac hypoxic condition is not primary aim of AAP recommended pulse oximetry in newborn. Page 5- Line 36. The authors state that the sensitivity of pulse oximetry is 42% and quote two papers, 7 and 8. As per paper 8 (Williams KB, Horst M, Hollinger EA, Freedman J, Demczko MM, Chowdhury D. Newborn Pulse Oximetry for Infants Born Out-of-Hospital. Pediatrics. 2021 Oct;148(4):e2020048785. The sensitivity of combined early and late screening was 66.7% (95% confidence interval [CI] 9.4% to 99.2%) for algorithm interpretation and 100% (95% CI 29.2% to 100%) for field interpretation. As per paper 7 (Jawin V, Ang HL, Omar A, Thong MK. Beyond Critical Congenital Heart Disease: Newborn Screening Using Pulse Oximetry for Neonatal Sepsis and Respiratory Diseases in a Middle-Income Country. PLoS One. 2015 Sep 11;10(9):e0137580.) The sensitivity and specificity of pulse oximetry screening for non-cardiac diseases were 42% and 99.9%, respectively, and 100% and 99.7% for CCHD, respectively. As per Cochrane, the overall sensitivity of pulse oximetry for the detection of CCHD was 76.3% (95% confidence interval [CI] 69.5 to 82.0) (low certainty of the evidence). Specificity was 99.9% (95% CI 99.7 to 99.9), with a false-positive rate of 0.14% (95% CI 0.07 to 0.22) (high certainty of the evidence). Plana MN, Zamora J, Suresh G, Fernandez-Pineda L, Thangaratinam S, Ewer AK. Pulse oximetry screening for critical congenital heart defects. Cochrane Database of Systematic Reviews 2018, Issue 3. Art. No.: CD011912. DOI: 10.1002/14651858.CD011912.pub2. Accessed 13 July 2023. The main goal for doing pulse oximetry, as per AAP recommendation, is to detect CCHD, and practically, it is called a CCHD screen by nurses. There is some doubt about the general statement of 42% pulse oximetry sensitivity, which may lead the reader to draw an incorrect conclusion. Authors should acknowledge Cochrane's meta-analysis and mention pulse oximetry's lower sensitivity for noncardiac conditions separately. It would be helpful if authors provided some insight into how machine learning tool will be used by community health workers who are minimally trained healthcare workers in low sociodemographic index countries if implemented nationally to reduce neonatal mortality rate.
--	---

REVIEWER	Dr. Sreehari Madhavankutty Nair Department of Health Services , Department of Health and Family Welfare
REVIEW RETURNED	23-Jul-2023

GENERAL COMMENTS	Adverse outcome reported at which point in time with regard to the age of the child is important, which is not found to be mentioned in the abstract. The sensitivity and specificity of both tools can be time sensitive (at which day of life it is done) and so it needs to be
---

	considered for better understanding and use of evidence in future. Adding POx to WHOS is found increasing the sensitivity, whether it is statistically significant and if so for which condition
--	--

VERSION 1 – AUTHOR RESPONSE

Dear Reviewer,

Thank you for your valuable comments on our manuscript. The comments have been addressed and all edits are in track changes in the revised manuscript with the responses included in the table below.

S. No	Comments by Reviewer 1	Author's Response
1.	I confine my remarks to statistical and methodological aspects of this paper. These were very well done. My only issue is that there needs to be more explanation of the figures regarding SHAP analysis.	We have expanded the legend of the corresponding figure to provide more explanation.
S. No	Comments by Reviewer 2	Author's Response
2.	In Table S3, I would like to know what the echocardiography findings were for the two patients who had false negatives test	The two false negative cases were Tetralogy of Fallot and isolated valvar pulmonary stenosis. This has been added in the text below table S3 in the supplementary material.
3.	The sensitivity and specificity of pulse oximetry for detecting neonatal sepsis and congenital heart disease are shown in Table S3. The oxygen saturation of neonates is affected by neonatal sepsis in the late stages. Pulse oximetry is not an effective tool for screening and early detection of neonatal sepsis. Consequently, it makes the impression that an excellent tool is being used to detect something that isn't intended, and its usefulness is being questioned.	Thank you for raising this concern. Hypoxia is seen in one third of children with neonatal sepsis (Swamy et al). This has been discussed in the papers by Jawen et al that have been referenced as no 7. In the current study, WHO signs and symptoms were used to detect sepsis and if positive was considered as a specific test for sepsis while pulse ox was considered as a specific test for CHD. Both the tools are modest in their sensitivity to pick its respective diseases that it is meant to screen for and that is why we are proposing a sequential ML model to improve the sensitivity of these highly specific tools and also proposing use of these tools to screen for at risk neonates.
4.	Abbreviating single words such as sensitivity (SE) and specificity (SP) does little to condense the text, but rather makes quick understanding difficult and train of thought bumpy.	We understand that this abbreviation may make the flow difficult for the reader. Based on your recommendation, we have used the full form of sensitivity and specificity throughout the text.
5.	I would like to know the method by which the study participants were selected. Was there	Thank you for raising this concern. Since this was not a trial, randomization was not

	any randomization done in order to prevent selection bias?	performed. A non-probability convenience sampling method was used to recruit participants. This has been added on page 6 of the manuscript.
6.	I have serious doubts about pulse oximetry's 42% sensitivity. Pulse oximetry sensitivity for diagnosing non-cardiac pathology should be clearly stated as diagnosing of non-cardiac hypoxic condition is not primary aim of AAP recommended pulse oximetry in newborn. As per paper 7 (Jawin V et al - The sensitivity and specificity of pulse oximetry screening for non-cardiac diseases were 42% and 99.9%, respectively, and 100% and 99.7% for CCHD, respectively. Page 5- Line 36. The authors state that the sensitivity of pulse oximetry is 42% and quote two papers, 7 and 8 .As per paper 8 (Williams KB et al -The sensitivity of combined early and late screening was 66.7% (95% confidence interval [CI] 9.4% to 99.2%) for algorithm interpretation and 100% (95% CI 29.2% to 100%) for field interpretation. As per Cochrane, the overall sensitivity of pulse oximetry for the detection of CCHD was 76.3% (95% confidence interval [CI] 69.5 to 82.0) (low certainty of the evidence). Specificity was 99.9% (95% CI 99.7 to 99.9), with a false-positive rate of 0.14% (95% CI 0.07 to 0.22) (high certainty of the evidence). Plana MN et al. The main goal for doing pulse oximetry, as per AAP recommendation, is to detect CCHD, and practically, it is called a CCHD screen by nurses. There is some doubt about the general statement of 42% pulse oximetry sensitivity, which may lead the reader to draw an incorrect conclusion. Authors should acknowledge Cochrane's meta-analysis and mention pulse oximetry's lower sensitivity for noncardiac conditions separately.	Thank you for your comment. As correctly pointed out that the 42% sensitivity is for "non-cardiac" conditions including sepsis. For clarity, we have added this on page 4 of the manuscript. As suggested, we have also added further details of the review by Plana et al on page 4 of the manuscript.
7.	It would be helpful if authors provided some insight into how machine learning tool will be used by community health workers who are minimally trained healthcare workers in low sociodemographic index countries if implemented nationally to reduce neonatal mortality rate.	Thank you for this important point. One possible application for this ML algorithm would be that the CHWs screen a child using pulse ox, followed by WHO signs and then have an application in their mobile phone which has the ML algorithm embedded. Entering the pulse ox, WHO and clinical information would help guide her as to the next steps of immediate referral, close follow up or well child. This has been

		described on page 15 of the manuscript
S. No	Comments by Reviewer 3	Author's Response
8.	Adverse outcome reported at which point in time with regard to the age of the child is important, which is not found to be mentioned in the abstract.	Adverse outcomes were assessed at 4 weeks of life. This is added in the abstract on page 2 of the manuscript.
9.	The sensitivity and specificity of both tools can be time sensitive (at which day of life it is done) and so it needs to be considered for better understanding and use of evidence in future.	We agree with the importance of timing of the pulse oximeter. As specified in table 1 on page 11 of the manuscript, 19% in < 24 hours, 34% in 24-48 hrs and 29% in 48-72 hrs.
10.	Adding POx to WHOS is found increasing the sensitivity, whether it is statistically significant and if so for which condition	Thank you for the comment. ML models are compared based on average performance and error bounds estimated over different random partitions of the datasets which have been described in Figure 4. One possibility is to run the McNemar test but because of the small number of true positives (<10), model assumptions would not be met. We also tried the paired Wilcoxon test on 6 independent partitions of the data. However, the small sample size of true positives when split in these partitions makes the model unstable. Keeping this in mind, we believe that this study adds to existing knowledge by reporting the possibility of better triage of at risk neonates when a sequential approach is used to screen them. A larger study is needed to prove its utility and this has been stated in the limitations section of the paper on page 15 of the manuscript.

Thank you for your valuable time in reviewing our submission.

VERSION 2 – REVIEW

REVIEWER	Dr. Imteyaz A. Khan Saint Peter's University Hospital, Pediatrics- NICU
REVIEW RETURNED	17-Sep-2023

GENERAL COMMENTS	I am grateful for the time you took to address my concerns.
---

REVIEWER	Dr. Peter Flom Peter Flom Consulting
-----------------	---

REVIEW RETURNED	18-Sep-2023
-------------

GENERAL COMMENTS	I had only very minor issues with the original version, and those have been addressed. I recommend publication.
---

VERSION 2 – AUTHOR RESPONSE

Not Applicable